# Travelers between cultures: The war and post-war generations

Marek Preiss[1,2,3] 🔾, Monika Fnaskova[3], Markéta Nečasova[3], Nikola Doubkova[1], Nikola Wolframova[3], Vojtěch Svoboda[3], David Ulcak[3], Edel Sanders[2] and Ivan Rektor[3,4]

[1]Department of Psychology, National Institute of Mental Health, Czechia, Czech Republic; [2]Department of Psychology, University of New York in Prague, Czech Republic; [3]Masaryk University, Central European Institute of Technology (CEITEC), Centre for Neuroscience, Czech Republic and [4]Department of Neurology, St. Anne's University Hospital and Faculty of Medicine, Czech Republic

## Research Article

**Keywords:**
trauma; Yugoslavia; resilience; adaptation; generations

**Corresponding author:**
Marek Preiss;
Email: marek.preiss@nudz.cz

## Abstract

War in the former Yugoslavia still reverberates in the lives of the generations that lived through it. The aim of this study was to compare a cohort that had direct experience of the war (first generation, G1, $n = 89$) with those born after the war (second generation, G2, $n = 30$). All participants stay or live in the Czech Republic. We used an individualized approach, with a structured interview of 91 questions, supplemented by quantitative methods to measure traumatic stress (PCL-5), adverse childhood experiences (ACEs) and centrality of the event (CES). G1 had a higher mean ACE score compared to G2, and the two generations did not differ in centrality of the event and trauma symptom severity, in the rate of psychiatric outpatient care use, psychiatric hospitalizations, diagnosed PTSD, current psychiatric medication use and in illicit drug use. A number of signs were indicative of good resilience, including the ability to move internationally, which implies language proficiency, and the ability to earn a sufficient income. G1 and G2 respondents represent a group of educated individuals with their mental health mostly matching that of the general population, as well as people who have success in their professional and personal lives.

## Impact statement

Are there differences in adapting to a different culture depending on whether people experience war or are born after it? Are there differences in the way they study, work and form relationships? The presented study reports that there are no significant differences, despite detailed face-to-face interviews with participants, combined with the use of questionnaire methods. The positive message of the study is that, despite the hardships of war, it is possible to live a happy life and adapt to the difficult task of moving to another country and culture.

## Introduction

The war in the former Yugoslavia, which began in 1991, significantly affected the lives of hundreds of thousands of people and forced the population to emigrate internally and externally. In the Bosnia and Herzegovina part of the territory alone, about 1 million people were displaced internally, and about 1.2 million escaped from the country (Šorgo and Živkovic, 2015; Barslund et al., 2017). The so-called Bosnian diaspora has spread to over 50 countries worldwide (Halilovich et al., 2018). During the war in the countries of the former Yugoslavia, more than 5000 people took refuge in the Czech Republic (Žižková, 2016), then after the war, most of them returned home. In other countries, there were more refugees from the former Yugoslavia (in 1992–1995, there were 320,000 registered refugees in Germany, 86,500 in Austria, 58,700 in Sweden, 22,000 in the Netherlands and 17,000 in Denmark; Barslund et al., 2017). The civilian population had been subjected to atrocities (UN, 1994) during the war. The most common traumatic experiences were shelling/bombing (85%), being expelled from home (38%) and learning about the murder of a cherished person (36%) (Nickerson et al., 2014). Not only men, but also women experienced being held in concentration camps or other kinds of detention centers (Dahl et al., 1998). Sexual violence was part of ethnic cleansing tactics (Zdrálková Grossová, 2021). Unsurprisingly, children during the war also suffered from the immediate direct and consequential effects of the war (Jones and Kafetsios, 2005).

The war ended after NATO intervention in 1995 (Šorgo and Živkovic, 2015), but continued later with the war in Kosovo (1999) and even later in Macedonia (2001). The NATO bombing of

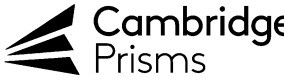



Serbian targets in 1999 was also another source of stress. The break-up of Yugoslavia resulted in the creation of six states. The war, through its impact on its participants, also affected the descendants of these participants on various levels, mainly social, for example, lifestyle (often impoverishment) and psychological (possible impact of traumatization).

The stress of the war in the former Yugoslavia affected the experience and behavior of the affected populations many years later. Also, several years after the war, people who stayed in the conflict area in addition to refugees suffered from paranoid ideation and anxiety (Priebe et al., 2013). High rates of PTSD have been found in emigrants from the region of the former Yugoslavia (Priebe et al., 2009), often continuing in the long term (Proroković et al., 2005; Comtesse et al., 2019). For example, migrants of the 1991–2001 Balkan wars to Sweden were associated with having a greater overall risk of being diagnosed with a psychiatric disorder (Thordardottir et al., 2020), highlighting the need for long-term monitoring of physical and psychological status. In interviews about the war in Iraq, children who had experienced the Balkan wars were more affected by the Iraq war than children from the United Kingdom, viewing the outcomes with more gravity and expressing a greater desire to end the war (Roberts et al., 2009).

According to Wasiak (2023), there is no doubt that the children of parents who experienced the war in the former Yugoslavia, and thus the collective trauma, "inherited" a troubled past. In contrast to the first generation, second-generation immigrants "find themselves caught between different and often competing generational, ideological, and moral reference points, including those of their parents, grandparents, and their own real and imagined perspectives on their multiple homelands" (Levitt, 2009, p. 1238). The second generation living in emigration also seeks a connection with the original (e.g., Bosnian) community (Maksimovic, 2023). Moreover, the risk of traumatization is inherent in the migration process itself (Szaló and Hamar, 2023).

Hypothetically, there are three stages of the offspring's coexistence with the parents' traumatic past: (1) in its mildest form, which burdens the self, it is a life with shadows of the past. (2) Living under their parents' shadows allows the past to negatively permeate the present and to manifest itself in a wrenching way. (3) When the traumatic past massively impedes the offspring's self-development, life takes place under the shadow of parental traumatization (Durban, 2011).

We know from research regarding experiences of World War II that the Holocaust experience includes a tendency not to talk about trauma, pointing to a conspiracy of silence (e.g., Danieli et al., 2016) similar to more recent studies of family narratives in Bosnia and Herzegovina. A study regarding the years between 2017 and 2021 examined Bosnian Muslim men aged 19–25 who were born during or just after the end of the war. It found that "silence has a protective function – both for parents and for 'protecting' children from painful memories of the war" (Wasiak, 2023, p. 82).

Questions regarding the adaptation of people from the same region, the former Yugoslavia, with and without direct war experience can be a useful addition to the study of traumatization and resilience. In this article, we are interested in comparing the first generation from former Yugoslavia, who personally experienced the war, and the second generation, children of parents from these countries, with no personal experience of the war and with the opportunity to temporarily live or to stay abroad in the Czech Republic.

## Sample

Eighty-nine participants from the first generation (G1) and 30 from the second generation (G2) were interviewed. The sample consisted of 23 blood relatives of the first generation cohort and 8 non-blood relatives (spouses).The recruitment of respondents for the study was done through personal contacts, social media and Lastavica (https://lastavica.org/) – a Prague-based association of citizens from the countries of the former Yugoslavia. Communication with all respondents was in the Czech language or in English.

### First generation (G1)

All respondents reported having direct experience with the Yugoslav Wars, a term that includes all the conflicts in the former Yugoslavia that occurred from March 1991 to August 2001. Respondents were aged 1–30 at the time the war began. Mean age was 38 years (SD = 8, range 24–71). There was a balanced gender ratio for the G1 members interviewed ($N = 89$) – 48% female and 52% male.

### Second generation (G2)

None of the respondents had direct experience of the war in the former Yugoslavia. The mean age was 24 years (SD = 2, range 21–28). G2 comprised 30 individuals (67% female) born between the years 1995 and 2002 to parents who had lived through the war in the former Yugoslavia.

## Methods

The study was conducted at the Central European Institute of Technology (CEITEC), Centre for Neuroscience at Masaryk University in Brno from 2022 to 2024. The research was approved by the ethics committee (EKV-2021-076) at Masaryk University and informed written consent was obtained from all participants.

For the purpose of the study, we designed a detailed interview supplemented by quantitative methods. The interview consisted of 91 questions and lasted approximately 60–90 min. It focused on three broad variables: (1) prewar experiences, (2) war experiences and (3) adaptation to the Czech Republic. Some questions were open-ended, whereas other items required response scales. For group G1, we asked closed questions, such as "Have you ever visited a psychiatrist in outpatient care?" (yes/no), and open questions, such as "Do you think your children would have a different personality if there had been no war?" We took a similar approach with G2, enabling both structured responses and the opportunity to express oneself openly. The answers were recorded using pen and paper and then transcribed into an electronic database.

Moreover, we measured the association of adverse childhood experiences (ACEs), the importance of challenging life situations to their identity and life (via the Centrality of Events Scale [CES]) and the level of traumatization (PTSD Checklist – PCL-5) using Spearman's correlation coefficient.

Participants were asked about basic demographic data, information regarding close people, events preceding the war, the course of experiences with the war in former Yugoslavia and psychological difficulties. In addition, they were asked to indicate their most stressful traumatic events, what effect the war had on them, their experience of living in the Czech Republic and their overall assessment of their own lives in both professional and personal spheres. The structured interview and psychological assessment methods

were administered by several authors of this article – M.N., M.N., N.W., V.S., and D.U.

### Adverse childhood experiences

The interview included questions on ACEs (Kalmakis and Chandler, 2014), to which the respondent answers yes/no. Administering questions on ACEs is quick, getting specific answers is relatively easy and implementing them into interviews that inform mental health professionals is simple. We have included an eight-question version (Bethell et al., 2017; Sacks and Murphey, 2018) about their experiences during the first 18 years of their lives. The total score ranges from 0 to 8. Higher scores represent a more intense rate of ACEs. Scores of 4 or more are considered to be clinically significant (Felitti et al., 1998).

### Centrality of events scale

The interview also included the CES (Berntsen and Rubin, 2006). This method measures how central an event is to a person's identity and life story. Questions were asked in connection with the most stressful life event participants experienced. The total score ranges from 0 to 35. A higher score on this scale represents a higher degree of event centrality. In the original study, an internal consistency of 0.88 was reported (Berntsen and Rubin, 2006).

### PTSD checklist for DSM-5

Furthermore, we administered the PCL-5 (Blevins et al., 2015). The PCL-5 is a 20-item self-report measure that assesses the 20 DSM-5 symptoms of PTSD. The total score ranges from 0 to 80. A higher score is associated with a greater level of PTSD symptoms. A cut-off raw score is 38 for a provisional diagnosis of PTSD and this cut-off has high sensitivity (0.78) and specificity (0.98) (Cohen et al., 2014).

### Data analysis

Statistical data processing (for data where comparisons were relevant – ACEs, CES, PTSD checklist – PCL-5, psychiatric care utilization data) was performed using Statistica version 14; $t$ test and Pearson's chi-square statistics were used to compare the two generations.

## Results

### First generation (G1)

#### Demographics

Participants from G1 experienced war as adults or as children. The year of birth ranged from 1952 to 1999. Most of the respondents were born in Sarajevo (15%); all but four respondents (96%) were born in the countries of the former Yugoslavia. When asked where they spent most of their childhood, the majority (53%) said Bosnia and Herzegovina (BiH). The most common educational backgrounds were university (53%) and high school (29%). The most common current nationalities selected were Bosnian (34%), Serbian (33%), Croatian (10%) and other.

#### Childhood adversities

The overall mean ACE score for G1 respondents was 2.21 (SD = 1.60, range 0–6). Twenty-one percent of the first-generation

cohort had scores of 4 or more, which are considered clinically significant (Felitti et al., 1998).

When we asked about respondents' adverse experiences in childhood (ACEs, first 18 years of life), we found that 15% of the cohort lived with a parent or caregiver who had divorced or separated from the family, 12% lived with a parent or caregiver who had died, 4% of respondents lived with a parent or caregiver who was in custody or prison, 20% lived as a child in a household with someone who was mentally ill, suicidal or severely depressed for more than a few weeks, 28% lived as a child with a person who had alcohol or drug problems, 26% lived with a parent, caregiver or other household member who was violent to others (slapped, hit, kicked, beat), 49% were victims of or observed violence in the neighborhood and 60% experienced material hardship rather often or very often.

### War events

When asked about major life stressors, 100% of G1 respondents reported war, sometimes in conjunction with migration, poverty, crime, hiding, military service, etc., and 68% reported stressors lasting for years. Respondents were aged 1–30 at the time the war began; 49% had experienced separation from loved ones due to the war (including as a result of emigration, and for women, the fathers' involvement in fighting), 89% had to go into hiding during the war, most often due to bombing, 4% suffered war injuries, 46% saw the dead or wounded and 3% experienced torture. War was identified as the most significant event by 49%. War still occurs as nightmares for 28%, and 2% said they were held hostage or captured during the war.

### Centrality of events scale

The mean of CES for G1 embedded in the survey was 24.3 (SD = 6.9, range 8–34); 36% of the respondents chose war as the central event in their lives, the rest chose various other situations, most often the death of parents, conflicts or break-ups.

For the description of the war events, respondents spoke about escaping to Prague in August 1992, emigration as a source of stress, fear for loved ones, bombing in 1999, hiding with family in a bunker in the basement, 4 months in exile with mother in Split then in a small village in BiH, hearing loud explosions, hiding in a cellar, being wounded by a grenade during the war resulting in shrapnel in the body, lack of food, hiding in a bunker with family, hiding in a shelter in a school, parents often being absent for work, hiding, especially the first month when they were particularly afraid – thereafter becoming numb (only in response to sirens), feeling terror and the murder of relatives.

### Coping and adapting

Sixty-one percent said they had come to terms with the war, 18% said they had "rather" come to terms with the war. When asked how the war had affected them, they reported, for example, that they valued life more than before, that resilience had increased and that they could cope better with stressful situations. Respondents who were children at the time of the war were more likely to report that they could not assess the impact of the war.

Forty-three percent of respondents have one or two children. Thirty-five percent of those with children reported that they do not talk to their children about the war at all, 17% reported that they talk to their children about the war and the rest gave various other answers. When asked about the impact of war on their children's upbringing, there were responses about being prepared for emergencies, increased caution, overprotection, among other responses.

### Psychological difficulties

Of the G1 cohort, 26% had sought outpatient psychiatric care at some point in their lives, 3% had been hospitalized for psychiatric care and 3% had been diagnosed as having PTSD. Eleven percent reported currently taking psychiatric medication. Experience with drug use was reported by 20%, and problems due to drinking alcohol sometime during life was reported by 13%.

### PTSD trauma symptom checklist – PCL-5

The mean score was 20.2 (SD = 13.4, range 0–54). Fourteen percent of the cohort had a score of 38 or higher, which is the cut-off raw score for a provisional diagnosis of PTSD (Cohen et al., 2014).

### Life in the Czech Republic

On average, participants started living in the Czech Republic at the age of 28 (SD = 7.5, range 11–47). Respondents (74%) reported that the language barrier in the Czech Republic was not experienced as a concern (47% reported "definitely not," 27% reported "rather not"). Having friends in the Czech Republic was mentioned by 88%; 33% said that they do not have a job corresponding to their education, 5% are still studying, 61% said they have a job corresponding to their education and 1% did not answer. Ninety-three percent said they were satisfied with life in the Czech Republic (54% reported "satisfied," 39% reported "rather satisfied"). Seventy-eight percent said they wanted to settle in the Czech Republic permanently or for a long time, 5% wanted to return to the country where they were born and 17% were considering another country or did not know.

When rating their income, 63% reported it as above average, 15% as below average and 11% as average (11% did not answer). Eighty-one percent said they were satisfied or somewhat satisfied with their job accomplishments, and 78% said they were satisfied or somewhat satisfied with their personal life accomplishments.

### National identity

When asked about subjective perceptions of national identity, 40 different definitions emerged. The most frequent were Bosnian (21%) and Serbian (15%), followed by specific identities, for example, not only combined Bosnian-Czech, Serbian-Czech, but also European, cosmopolitan, Dalmatian, international, human, "under construction," etc.

## Second generation (G2)

### Demographics

Ninety-five percent of the second-generation cohort were born in former Yugoslavia, and 5% were born abroad. Sixty-three percent spent most of their childhood in BiH, 16% in the Czech Republic and the rest in various other countries, for example, Croatia (13%), Cyprus, Turkey and Bosnia. The most common nationalities were Bosnian and Herzegovinian (47%), Croatian (17%), Serbian (13%) and Czech (13%). Seventeen were born in the Czech Republic. The majority of respondents started living in the Czech Republic between the years 2021 and 2023 (77%), for example, for academic pursuits or for other reasons. Most respondents (70%) are studying or have graduated from a university, 70% live in big cities (Prague, Brno) and 100% are single.

### Childhood adversities

The mean ACEs for G1 was 1.53 (SD = 1.46, range 0–4). Thirteen percent of the second-generation cohort had scores of 4 or more, which are considered clinically significant (Felitti et al., 1998) as

stated before. (Note that 21% of the first-generation cohort had scores of 4 or more.)

When we asked about respondents' experiences in childhood (ACEs, first 18 years of life), we found that 27% of the cohort lived with a parent or caregiver who had divorced or separated from the family, 13% lived with a parent or caregiver who had died, 7% of respondents lived with a parent or caregiver who was in custody or prison, 20% lived as a child in a household with someone who was mentally ill, suicidal or severely depressed for more than a few weeks, 20% lived as a child with a person who had alcohol or drug problems, 20% lived with a parent, caregiver or other household member who was violent to others (slapped, hit, kicked, beat), 20% were victims of or observed violence in the neighborhood and 23% experienced material hardship rather often or very often.

### War events

Respondents were aware of their parents' wartime experiences and mentioned a number of them (including hiding, injuries, escapes, exposure to life-threatening situations and parents' having psychological difficulties). Thirty-three percent mentioned their parents' war injuries, 63% their direct experience of combat, 33% war-related deaths in the family and 43% mentioned deaths of neighbors as a result of the war.

### Centrality of events scale

The mean of CES in this group was 22.3 (SD = 6.5, range 11–35) versus 24.3 in the first-generation group. Examples of topics selected included separation from family during COVID-19, father's cancer, grandmother's illness, parents' divorce, loss of a loved one, childhood violence, father's suicide attempt, etc.

### Psychological difficulties

Seventeen percent of the G2 cohort reported outpatient contact with a psychiatrist, 3% reported psychiatric hospitalization and 7% reported currently taking psychiatric medication. Experience with drug use was reported by 13%, problem drinking of alcohol sometime during their life was reported by 3%. None of the respondents had been diagnosed as having PTSD, though one said he thought he might suffer from it.

### PTSD trauma symptom checklist – PCL-5

The mean score of the G2 cohort was 20.9 (SD = 14.1; range 2–58), and 12% of the cohort had a score of 38 or higher, which as specified earlier, is the cut-off raw score for a provisional diagnosis of PTSD (Cohen et al., 2014). (Note that the mean score of the G1 cohort was 20.2, and 14% had a score of 38 or higher.)

### Communications with parents

Fifty-three percent said that the war had affected how they were brought up, with over-protection of offspring, increased vigilance, pride and preparedness for unexpected situations recurring such as in the examples given. Sixty-three percent said that their parents communicated with them about the war, and 84% said "yes" or "rather yes" when asked whether the war had affected them even though they had not experienced it directly. When asked how the war had affected them, they reported, among other things, that they were prepared for having to survive, that they felt sadness, that they were generally more afraid and that they felt more stressed. When asked whether they thought they would be different in character if there had been no war, 66% said "yes" or "more likely yes." When asked whether the war had affected the way they interacted with other people, 63% said "yes," citing both positives and negatives.

### Life in the Czech Republic

Respondents (53%) reported that the language barrier in the Czech Republic was not experienced as a concern (definitely not by 20%, rather not by 33%). Sixty-three percent said they have friends in the Czech Republic, and 83% said they are satisfied with their life in the Czech Republic – (30%) satisfied and (53%) rather satisfied. When assessing their income, 30% were unable to rate it because they were not yet working, 37% reported it as below average, 27% as average and 6% as above average. Seventy percent reported being satisfied or somewhat satisfied with their career successes, and 73% reported being satisfied or somewhat satisfied with their personal life successes.

### National identity

When asked about subjective perceptions of national identity, 15 different definitions emerged: Bosnian (33%) and Serbian, Yugoslav and Croatian (10% – the latter three combined) were the most common national identities reported, followed by specific identities such as combined Bosnian–European, European, nomad and Balkan. Czech did not appear either separately or in combination with another identity.

### *Comparison of G1 and G2*

#### Demographics

Both G1 and G2 had a relatively high educational attainment, with 53% of G1 having a university degree and 70% of G2. Except for four respondents in G1, the rest of that cohort (96%) were born in the former Yugoslavia; in G2, 56% were born in BiH and the rest in other countries.

#### Childhood adversities

G1 had a higher mean ACE score compared to G2 ($t(115) = -2.232$, $p = 0.044$). For two items, G1 had higher values; more often they (49% vs. 20%) were victims of or they observed violence in the neighborhood ($\chi^2(1) = 6.732$, $p = 0.009$) and more often (60% vs. 23%) they experienced material hardship rather often or very often ($\chi^2(1) = 13.098$, $p < 0.001$).

#### Centrality of events scale

The difference between groups in the CES total scores was not statistically significant ($t(113) = -1.248$, $p = 0.214$).

#### Psychological difficulties

The two generations did not differ in the rate of psychiatric outpatient care use ($\chi^2(2) = 2.925$, $p = 0.116$), psychiatric hospitalizations ($\chi^2(2) = 4.307$, $p = 0.232$), diagnosed PTSD ($\chi^2(2) = 3.935$, $p = 0.139$), current psychiatric medication use ($\chi^2(2) = 3.222$, $p = 0.199$) or in drug use ($\chi^2(3) = 6.916$, $p = 0.746$).

#### Rate of traumatic symptoms: PTSD checklist – PCL-5

The difference between both groups in the PCL-5 measure of traumatic symptoms was not statistically significant ($t(115) = -0.226$, $p = 0.821$).

### Discussion

A total of 129 respondents who either experienced the war or were born after it participated in a structured interview and measures via quantitative methods. The structured interview focused on adaptation, traumatization and resilience, while the quantitative methods complemented the qualitative information in the areas of adverse childhood events, centrality of events and posttraumatic symptoms. Most of the information we obtained supports the notion that respondents adapted well to migration and relocation experiences.

All G1 respondents had experienced war in the countries of the former Yugoslavia; all G2 respondents had no direct experience of war. The majority of both G1 and G2 had retained their nationality and had not become citizens of the Czech Republic. Only 17% of G2 were born in the Czech Republic, and the majority started living in the Czech Republic between the years 2021 and 2023 (77%).

G1 had a higher rate of child adversities compared to G2. Yet, G2 was more likely to live in families in which parents or caregivers had separated (27% vs. 15%). The largest differences between generations were in the issue of violence (regarding the question about whether they were victims of or had observed violence in the neighborhood), which was at 49% for G1 and 20% for G2, and in the experience of material hardship during childhood, which was at 60% for G1 and 23% for G2.

Conversely, G2 was more likely than G1 (27% vs. 15%) to report that a parent or caregiver had divorced or separated from the family. Differences represent differences in the formative family milieu; while G1 experienced more childhood violence and material deprivation (both at least partially related to the war), G2 experienced more parental separation, which may or may not be related to the effects of the war. However, compared to data from peacetime conditions in the Czech Republic, these numbers for G2 in terms of parental separation are still relatively low, as parental separation or divorce is 23% and parental violence is 22.1% (Velemínský et al., 2020). As has been previously recognized, research is needed to find whether interventions such as parent training programs might mitigate the intergenerational transmission of trauma (Zhang et al., 2023).

The first generation reported personal experiences of the war, and the war was the most significant life experience for 49% of respondents, but when rated on the CES scale, "only" 36% reported the war as a perceived major event. Thus, it seems that even with a time gap of more than 20 years (depending on whether one considers the end of the war to be 1995 with the Dayton Accords or 2001 after the end of other conflicts in the region), the war is alive in the mind and constitutes a key life event for approximately one third to one half of respondents.

However, it seems that despite the severity of the war in their experience, the respondents in G1 who have children of their own do not communicate with them about the war – 35% of G1 respondents who have children stated that they do not talk to them about the war at all, which may correspond to the finding that "it is certain that in Bosniaks' homes, there was, and still is, a lack of space for honest conversations about the war" (Wasiak, 2023, p. 81). The "conspiracy of silence" seems to be true for some of the respondents, though the reasons for silence may be different and our data do not allow us to reveal whether the silence is caused by the horror of the experience, shame and guilt, desire to protect their children from disturbing information or other reasons. More than half (53%) of G2 respondents reported the effects of the war on their parents, which were reflected in their own upbringing. Further, 84% of G2 reported that the war had affected them even though they had not experienced it themselves, indicating the existence of and potentially an awareness of the transgenerational transmission of trauma as well. In other words, in the minds of G2 respondents, there appears to be an awareness that the war is also imprinted in their life story and can influence one's attitudes, decisions and life

course in a similar way to other groups going through particularly challenging life situations, such as expulsion from the country in the case of dissidents in former Czechoslovakia (Heissler et al., 2024) or current refugees from Ukraine (Preiss et al., 2024).

Given that those living in the territory of the former Yugoslavia were found to have "a lack of space for honest conversations about the war" (Wasiak, 2023, p. 79), the greater openness of respondents in this study to communicate about the war can perhaps be attributed to the situation of emigration and travel between cultures. Although we expected G1's war experiences to influence overall higher scores on the Centrality of Events Scale (CES), the differences between G1 and G2 were not statistically significant even though the G2 had lower scores than the G1. For G1, the war was selected as the most significant stressful life event; for G2, the selected events were varied and corresponded to common peacetime concerns. In a previous study of Holocaust survivors and their descendants, event centrality was positively associated across generations in Holocaust G2 and G3, and Holocaust G1 PTSD symptoms predicted Holocaust G3 event centrality, but a modified methodology for measuring event centrality was used (Greenblatt-Kimron et al., 2021). Although G1 respondents most often selected war as the event and G2 respondents selected other events, the level of centrality did not differ, just as the level of posttraumatic symptoms did not differ between G1 and G2. Since event centrality refers to the degree to which a traumatic event becomes a reference point for the interpretation of everyday suppositions, it can be concluded that the war in the former Yugoslavia did not leave such a powerful mental scar in our respondents as in the case of the Holocaust.

We might expect G1 to have, given the wartime experiences, a higher level of posttraumatic symptoms. However, the difference in posttraumatic stress symptoms was not significant, and the percentage of G1 and G2 individuals with probable PTSD based on self-assessment was comparable, 14% and 12%, respectively, at a cut-off raw score of 38 (PTSD checklist – PCL-5; Blevins et al., 2015). The reported rate of clinically diagnosed PTSD was 3% for G1 and 0% for G2. Several possible mechanisms of trauma transmission have been identified for asylum-seeking and refugee families – maladaptive parenting styles, reduced emotional availability of parents, reduced family functioning, accumulation of family stressors, dysfunctional intrafamily communication styles and severity of parental symptomatology (Flanagan et al., 2020). Based on the information we have on G1 and G2 respondents, none of these mechanisms were significantly present, though not all were examined in detail. On the other hand, we know that collectively traumatized people exhibit symptoms that go beyond the PTSD criteria observable at the supraindividual level (Mutuyimana and Maercker, 2023).

In the United States, the lifetime prevalence of PTSD is from 3.4% to 26.9% (Schein et al., 2021), with higher prevalence in the military-experienced population according to the data cited in this review. Our reported values based on PCL-5 results are approximately in the middle of this range, but this is a self-assessment that may not correspond to a clinical diagnosis. Our data contrast with findings of high prevalence of PTSD approximately 10 years after the onset of war trauma (84% of participants; Priebe et al., 2009), persistence of depression and PTSD over time (Mollica et al., 2001) and paranoid ideation and anxiety (Priebe et al., 2013).

The reason for the relatively low rate of PTSD may be good resilience due to, among other things, (1) education (in G1, 82% have completed secondary school or university; in G2, 70% have studied or are studying at the higher education level) as well as

(2) the ability to move internationally, which implies language proficiency, together with the ability to earn a sufficient income (in both G1 and G2, 62% reported above-average incomes). Variables associated with less favorable post-war mental health impacts include older age, female gender, lower education, shorter time since conflict, higher trauma exposure, displacement, limited economic opportunities and ongoing political conflict (Porter and Haslam, 2001) as well as lack of acceptance in the host country (Bogic et al., 2012).

However, our G1 respondents were relatively young (the average age was 38 years at the time of the study), the sample had a balanced proportion of males and females (52% and 48%), the time since the war was relatively long (data collection was during 2022–2024), our respondents were in good economic circumstances, lived in peaceful conditions and were in contact with their country of origin, which is only a few hundred kilometers away. Use of mental health care was more frequent in G1 compared to G2 (26% vs. 17%), but the difference was not statistically significant. We found no differences between generations in the rate of psychiatric outpatient care use, psychiatric hospitalizations, current psychiatric medication use and in drug use.

The majority (74%) of G1 respondents has mastered the initial language barrier in the Czech Republic, the majority (88%) have friends in the Czech Republic and the majority (93%) express satisfaction with life in the country. It seems that these data can be interpreted as revealing good adaptation to different cultural conditions.

Respondents from G1 created 40 different definitions of national identity; G2 created 15. For G1, there were different combinations of nationalities. Czech national identity was combined with others, national identity associated with only one state was in the minority (36%) and Czech identity alone was not present. For G2, as with G1, Czech identity alone was not present at all, despite the fact that 16% of G2 respondents were born in the Czech Republic and 13% of G2 had Czech nationality. Thus, for the majority of G2 respondents, the Czech Republic is the place where they live and study, but not the place that is a threat to their national identity. This finding is consistent with another finding from the Swiss diaspora "in the process of figuring out their relationship to the ancestral homeland, second generation Bosnians develop and articulate various, often diverging, notions of belonging" (Müller-Suleymanova, 2023, p. 1798) and the finding that "the position of the second generation becomes characterized by a higher degree of ambivalence towards the self, the other and the past while they still gain their identity from this past" (Yordanova, 2015, p.85). At the same time, it is evident that the national identity of G2 is more stable compared to G1, which faced difficult life choices in the past – whether to stay or to leave.

We found that psychological differences are minimal, despite very different experiences related to war. Based on our data alone, we could not determine whether this result is due to resilience factors, cultural adaptation or other mediating variables affecting the mental health of both generations. We are inclined to believe that our sample represents educated, internationally mobile individuals who do not represent the population of the former Yugoslavia, but rather a part of it, representing the creative, dynamic, active and resourceful part of its population, a group that knows how to cope in times of difficulty. This is also documented by the creative activity of this first generation (G1) in emigration, for example, Medenčević (2021) and Jaganjac (2014). Only in the future will it be possible to assess whether G2 will be similarly creative.

## Limitations

Although we are writing about two generations, war and post-war, the average age difference between them is relatively small – 14 years. Therefore, they do not form a generation in the true sense of the word. Given the small age difference between the two groups (14 years), it is difficult to interpret similarities or differences in terms of generational differences. It is possible that increasing the age difference could lead to different results (despite our best efforts, however, we were unable to find enough participants).

The individuals who comprise G2, for the most part, are not the direct descendants of the individuals in G1 who participated in this study; rather, they are primarily of a different cohort. Furthermore, G2 had a smaller number of respondents than G1, and a different male/female ratio, which may have biased the results. Although appropriate and efficient for this study, structured interview and questionnaire methods do not allow the richness of data that qualitative methods such as life story or clinical methods aimed at making a diagnosis can achieve.

## Conclusions

It can be concluded that both G1 and G2 respondents represent a group of educated, internationally mobile individuals. The typical effects of war – PTSD and symptoms of traumatic stress – observed in the first and sometimes in subsequent generations of this and other studies, were not found in this study to a notable degree. The mental health of respondents in our study largely matches that of the general population. They have success in their professional and personal life. They show signs of resilience and vitality, and can be an example of adaptively embracing a difficult life situation – the experience of war and uprooting. For future studies, a closer evaluation of resiliency factors revealed by the respondents in this study could be utilized to help other populations who also had experienced traumatic circumstances to adapt and grow in a similarly positive direction.

**Open peer review.** To view the open peer review materials for this article, please visit http://doi.org/10.1017/gmh.2025.10085.

**Data availability statement.** The data supporting the findings of this study are available upon reasonable request to the corresponding author.

**Author contribution.** M.P.: investigation, resources, data curation, writing original draft, visualization. M.F.: project administration, investigation, resources, data curation, writing original draft, visualization, investigation. M.N.: investigation, data collection, writing original draft. N.D.: writing original draft, visualization, investigation. N.W., V.S.: investigation, data collection. D.U.: investigation, visualization. E.S.: writing original draft, proof reading, statistical analysis. I.R.: conceptualization, supervision.

**Financial support.** Supported by the Ministry of Health of the Czech Republic in cooperation with the Czech Health Research Council under project no. NU22-04-00661.

**Competing interests.** The authors declare no potential conflicts of interest with respect to the research, authorship and/or publication of this article.

**Ethics approval statement.** Ethical approval for this study was obtained from the ethics committee of the Masaryk University (approval code EKV-2021-076) on June 24, 2021.

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
