## [Reviewer Report]

The study provides valuable insights into the mental health, adaptation, and resilience of two generations affected by the war in former Yugoslavia. The sample consists of educated individuals with positive experiences of migration and relocation. The findings suggest that both generations display resilience and have adapted well to their new environments, showing no significant differences in mental health outcomes such as PTSD. One of the strengths of this study is its focus on both the direct experiences of war (G1) and the transgenerational transmission of trauma (G2), allowing for a nuanced understanding of the impact of conflict across generations. Moreover, the respondents’ high level of education and positive economic outcomes contribute to their good adaptation and psychological well-being.

However, there are certain aspects of the study that could be improved. First, while the study provides a comprehensive overview of the respondents' mental health, the differences between G1 and G2 could be more thoroughly explained, especially in relation to mental health outcomes. The fact that both generations display minimal differences in psychological health, despite the significant contrast in their experiences, warrants further analysis. A deeper exploration of why psychological differences are minimal, despite the vastly different experiences related to the war, could offer valuable insights. It would be beneficial to examine whether this result is due to resilience factors, cultural adaptation, or other mediating variables influencing the mental health of both generations.

In addition, the authors could benefit from expanding the discussion on mental health disparities between the two groups. Exploring whether there are specific psychological mechanisms that allow G2 (the post-war generation) to exhibit resilience despite not experiencing war directly would contribute significantly to the overall understanding of transgenerational trauma. Additionally, exploring more deeply the limitations of the sample, including the small age difference between G1 and G2, would enhance the study’s robustness. Given the relatively small age gap, this might limit the interpretation of the generational differences and their impacts.

Finally, for future research, it is recommended that a more diverse sample, with a larger representation of both genders and a broader age range, be used. Further exploration into how cultural identity and the migration experience contribute to mental health resilience would provide a better understanding of the long-term psychological impacts of displacement.

The paper is accepted, but it is recommended that, in accordance with the suggestions provided, efforts be made to improve certain sections of the work to enhance its quality.

---

## [Reviewer Report]

I evaluate the article positively. The conducted study was carried out meticulously, with a clear awareness of its limitations, which reflects the authors’ scientific diligence. The topic addressed is of great cognitive and social relevance—especially in the context of today’s geopolitical realities, where the experiences of individuals affected by war trauma are becoming increasingly common. Research of this kind is extremely important, as it can contribute to the development of effective support systems based on knowledge drawn from previous experiences and analyses.

I strongly recommend this article for publication.

---

## [Reviewer Report]

Thank you for this interesting manuscript. It reports on a study that compared first-generation individuals who had experienced the wars in the former Yugoslavia and second-generation individuals whose parents had direct exposure. Both groups were residing in the Czech Republic. The authors are to be commended for their focus on stress-related symptoms, as well as resilience and adaptation-related aspects. However, there are several major concerns.

Abstract: Some information is missing: What are the group sizes? Is the research being conducted in the Czech Republic? How was the data analyzed? What questions did the interview comprise, which variables were extracted?

Introduction: There is a large body of literature on the intergenerational transmission of trauma effects (e.g., https://doi.org/10.1002/cpp.2836, https://doi.org/10.1080/20008198.2020.1790283,

https://doi.org/10.1177/15248380221126186) that the authors may cite in order to explain why and how the second generation might be affected. This could also justify variable selection, for example, event centrality could be a mechanism of transmission (e.g., DOI: 10.1016/j.janxdis.2021.102401).

Method: Based on the information provided, the study could not be replicated. The authors should provide more detail on their exact procedures and the interviews they conducted. Are some members of the two groups related or these independent samples? Who conducted the interviews? How was the data extracted and analyzed? Did it include open questions? If so, are there central statements to provide as examples? What variables were derived from it? These details could be included in a supplement.

Discussion: It is more a repetition of the results. However, the findings should be discussed with regard to the central literature on intergenerational trauma transmission and in relation to migrant adaptation. Also, I can see several limitations to this study (e.g., generalizability, small sample size) and they authors might include them.

---

## [Reviewer Report]

The authors addressed my suggestions by expanding the Discussion and further elaborating on the psychological mechanisms of resilience, as well as including sample limitations and directions for future research. These revisions enhance the clarity and relevance of the manuscript, although some issues could be explored further in future versions. Overall, the manuscript shows improvement compared to the previous version, and I recommend it for publication.